# Griddient: a microfluidic array to generate reconfigurable gradients on-demand for spatial biology applications

Cristina Sanchez-de-Diego[1,2,3,9], María Virumbrales-Muñoz [1,2,3,4,9], Brock Hermes [1,9], Terry D. Juang[1,2,3], Duane S. Juang[1,2,3], Jeremiah Riendeau [5], Emmanuel Contreras Guzman [5], Catherine A. Reed-McBain [2,3,6], Sara Abizanda-Campo[2,3,6], Janmesh Patel [3,6,7], Nicholas J. Hess[3,8], Melissa C. Skala [2,3,5], David J. Beebe [1,2,3] & Jose M. Ayuso [2,3,6 ✉]

Biological tissues are highly organized structures where spatial-temporal gradients (e.g., nutrients, hypoxia, cytokines) modulate multiple physiological and pathological processes including inflammation, tissue regeneration, embryogenesis, and cancer progression. Current in vitro technologies struggle to capture the complexity of these transient microenvironmental gradients, do not provide dynamic control over the gradient profile, are complex and poorly suited for high throughput applications. Therefore, we have designed Griddent, a user-friendly platform with the capability of generating controllable and reversible gradients in a 3D microenvironment. Our platform consists of an array of 32 microfluidic chambers connected to a 384 well-array through a diffusion port at the bottom of each reservoir well. The diffusion ports are optimized to ensure gradient stability and facilitate manual micropipette loading. This platform is compatible with molecular and functional spatial biology as well as optical and fluorescence microscopy. In this work, we have used this platform to study cancer progression.

[1] Department of Pathology & Laboratory Medicine, University of Wisconsin, Madison, WI, USA. [2] Department of Biomedical Engineering, University of Wisconsin, Madison, WI, USA. [3] The University of Wisconsin Carbone Cancer Center, University of Wisconsin, Madison, WI, USA. [4] Department of Cell and Regenerative Biology, University of Wisconsin, Madison, WI, USA. [5] Morgridge Institute for Research, 330 NOrchard street, Madison, WI, USA. [6] Department of Dermatology, University of Wisconsin, Madison, WI, USA. [7] University of Wisconsin School of Medicine and Public Health, University of Wisconsin, Madison, WI, USA. [8] Department of Medicine, Division of Hematology, Medical Oncology and Palliative Care, Madison, WI, USA. [9]These authors contributed equally: Cristina Sanchez-de-Diego, María Virumbrales-Muñoz, Brock Hermes. ✉email: ayusodomingu@wisc.edu

Solid tumors are complex systems where gradients of nutrients, oxygen, and waste products play a crucial role in multiple steps of tumor development and drug response[1]. In the tumor microenvironment (TME), accelerated tumor cell proliferation commonly leads to nutrient consumption, waste product accumulation, acidic pH, and hypoxia within the tumor mass[2,3]. Additionally, these nutrient gradients evolve over time as the tumor grows or new blood vessels reach the tumor, creating a complex environment that changes spatially and temporally[4]. Tumor cells must undergo extensive changes including metabolic and genetic reprogramming to survive amidst the changing and nutrient-depleted TME[5]. These cellular and metabolic adaptations can include an accelerated autophagy, overexpression of genes involved in pH regulation (e.g., carbonic anhydrase 9), modulation of proliferation rate, or increased migration[6–8].

Furthermore, recent reports have demonstrated that cyclic exposure to hypoxia and nutrient starvation activates compensatory mechanisms that increase tumor aggressiveness once nutrient supply is restored (e.g.,metastatic potential)[8–10]. Finally, nutrient starvation severely compromises the capacity of the immune system to destroy tumor cells. Effector cells such as T and natural killer (NK) cells are rapidly exhausted and lose their cytotoxic capacity as they are exposed to cyclic starvation[11,12]. This exhausted phenotype is not reversible once nutrients are replenished, crippling the capacity of these nutrient-starved immune cells to prevent tumor growth[11].

Despite previous research, the molecular pathways driving tumor adaptation and immune exhaustion are not completely understood. However, capturing the complex and evolving TME with traditional Petri dishes remains challenging. Numerous reports have demonstrated the potential of microfluidic devices to generate biochemical gradients to study cell response[11,13–15]. We report a microfluidic platform that allows the generation of transient spatial and temporal gradients on-demand. Our microfluidic platform leverages capillary forces to generate reconfigurable gradients in a simplistic, yet robust manner. demonstrated Following validation of the device's ability to generate complex gradients, we showcased its capacity to selectively recover cells from different locations of the array to evaluate tumor metabolic adaptation or to generate immunotherapy grafts to explore further cellular treatments.

## Results

### Microdevice operation
In this work, we designed a microfluidic device called Griddient to circumvent these limitations. The Griddient platform consisted of an array of 32 microfluidic chambers for 3D culture connected, wherein each chamber was linked to an array of reservoir wells following the 384 well-plate design (Fig. 1a–c and Supplementary Fig. 1). Diffusion ports located at the bottom of each well (Ø 1.7 mm) facilitate connectivity between the chambers and the reservoir wells. These diffusion ports have been optimized for manual micropipette loading to ensure micropipette tips fit properly while simultaneously forming a capillary valve that facilitates liquid confinement in the chamber (Supplementary Fig. 1). A cell suspension in a collagen hydrogel can be injected into the main chamber through one of the diffusion ports (Fig. 1d). After 15 min at room temperature, the collagen was fully polymerized and endothelial cells were seeded as a monolayer on top of the interface between the collagen hydrogel and the bottom of the reservoir well (Fig. 1d), mimicking the endothelial layer. Finally, media or PBS were added to the reservoir wells, which allowed us to manipulate nutrient concentration throughout the platform (Fig. 1d).

To test the application of the design to generate and manipulate complex gradients we pipetted solutions containing food dye in the reservoir wells. We observed that the platform can contain the media properly (Fig. 1b). Later we used fluorescently labelled dextrans (i.e., 40 kDa FITC-dextran and 70 kDa Texas Red- dextran) and demonstrate the capacity of Griddient to create different configurations including linear, radial, or periodic gradients (Fig. 1e–g). We monitored the dextran diffusion for 48 h and analyzed the diffusion patter in between the wells. Dextran started to diffuse after 3 h of incubations and keep diffusing over the time. After 15 h of incubations, the gradient reached to a stable state in which reminded for at least 48 h. (Fig. 2). We also observed that big differences in volume affect the diffusion patter and can alter the gradient (Suplementary Fig. 2)

To demonstrate the capacity of Griddient to support the cell growth, we seeded HCT-116 cells at high density (30 million/mL) in the collagen gel. After collagen polymerization, media was added to all the reservoir wells. Diffusion ports were added to ensure that nutrients diffused homogenously across the chamber, as demonstrated by the high viability of the cells underneath and between reservoir wells (>95% of cell viability after seeding) (Fig. 3). We maintained these cultures in the incubator for 3 h (0 days), 7, 15, and 30 days to monitor cell viability in the Griddient. Additionally, we did not observe the generation of low viability domains in the space between reservoir wells, suggesting that nutrient passive diffusion was fast enough to support cell viability even in regions located in between wells. Moreover, we did not observe the formation of spatial heterogeneities, or collagen regression over the 30 days of the experiment, suggesting that the Griddient device is a stable platform for long-term experiment.

### Generation of controlled metabolic microenvironments in the Griddient device
Several human pathologies, such as cancer or stroke, are characterized by limited blood flow, which generates nutrient starvation and waste product accumulation. The combination of these factors influences phenotype and gene expression across the tissue. We assessed the capacity of the Griddient to generate nutrient gradient configurations and their effect on cell viability. We seeded HCT-116 cells/mL at high density in the microfluidic chambers of the Griddient. After gel polymerization, the reservoir wells were filled with culture medium or PBS following the desired spatial configuration (Fig. 4). We cultured the cells in these conditions for 24 h and then investigated cell viability by staining total and dead cells with CellTracker™ Green CMFDA and propidium iodide (PI) respectively. When media was added only to the leftmost wells, we were able to generate a necrotic front where most live cells were located in the left flank of the chamber (Fig. 4a). Interestingly, we observed that cells cultured underneath the first column of PBS wells still showed high viability compared with the other PBS wells. These results suggested that nutrient diffusion alone was enough to maintain cell viability under the PBS wells that were adjacent to media wells. When we pipetted media in all the wells but the two central ones (Fig. 4b), we observed the generation of a central area of low cell viability. As in the previous configuration, the cells where media was added and in the interface between the media and the PBS remained viable. When media was added only to the central wells, we observed that cells underneath the media remained viable, and the diffusing media was sufficient to maintain the viability of the cells in the interface between the media and the PBS while wells with PBS showed significantly higher cell death (Fig. 4c).

Cancer is often treated with chemotherapeutic drugs. In the case of solid tumors, these drugs must penetrate the tumor, creating a gradient of concentration that is challenging to reproduce in vitro. Griddient also allows us to generate and manipulate drug gradients within the microfluidic chamber; thus,

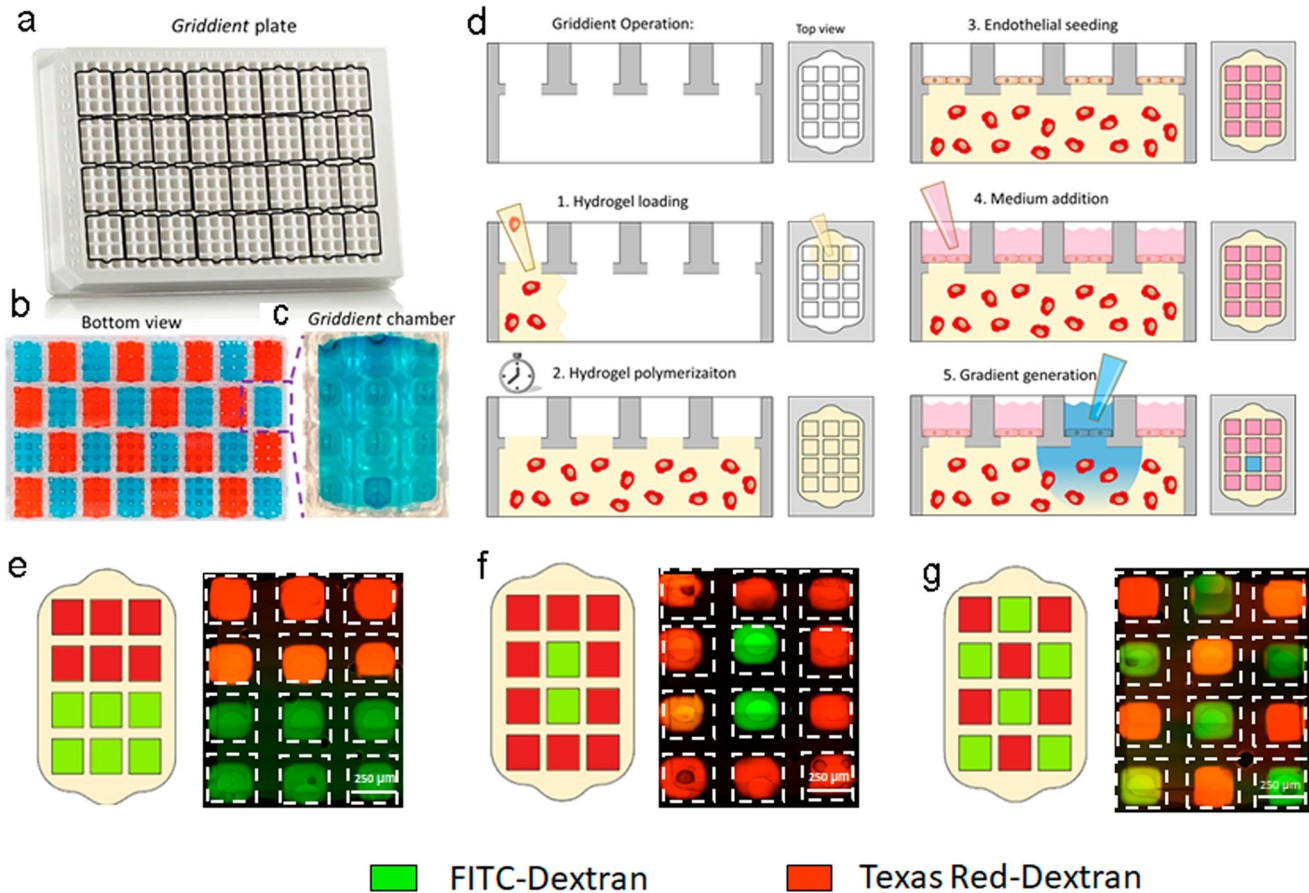

**Fig. 1 Griddient operation. a** Image of the Griddient platform, black outlines show the 32 individual microfluidic chambers underneath the platform. **b**, **c** Picture of the bottom of the Griddient platform showing the microfluidic chambers filled with food dye for visualization purposes. **d** Experimental protocol for the Griddient operation. 1. Cells were mixed with a type 1 rat tail collagen solution and injected into the main chamber through one of the diffusion ports. 2. Collagen polymerized for 10–15 min at room temperature. 3. When the collagen is fully polymerized, endothelial cells are seeded on top of the hydrogel interface as a monolayer. 4. After endothelial cells have been attached, media or PBS, were added to the reservoir wells. 5. The media/PBS of each well can be modified to generate or revert the desired gradients. E-G. 40 kDa FITC-Dextran and 70 kDa TR-dextran diffusion in the Griddient after 24 h in lineal (**e**), radial (**f**), or periodic (**g**) configuration.

enabling us to explore cell response to sub-lethal drug concentration, which often lead to drug resistance. To demonstrate this capability, we evaluated the efficacy and diffusion of puromycin, a protein synthesis inhibitor. We seeded HCT-116 cells at high density in the platform and added 5 mg/mL of puromycin to the two central wells. After 24 h of treatment, we observed the generation of a necrotic core corresponding to the areas of higher puromycin concentration. The diffusion of puromycin induced cell death with a greater than 3 mm radius from the center of the well where it was applied (Fig. 4d).

Spatial transcriptomics and other emerging imaging techniques are showing that solid tumors, and human tissue in general, are highly complex structures with heterogeneous spatial organization. Nutrient availability, and other biochemical gradients, exert strong evolutionary pressure on solid tumors; which, in turn, leads to the generation of subdomains within the tumor mass with critical differences in their genetic, epigenetic, or metabolic profile. One of the main limitations of current techniques used in spatial profiling is their limited potential to selectively retrieve live cells from these spatial sub-domains for downstream cellular analysis. Commonly, these techniques require the use of destructive protocols (e.g., tissue fixation, cell lysis) that prevent the isolation of live cells. Thus, we designed the Griddient to allow the isolation of viable cells for downstream molecular and even functional profiling. The Griddient diffusion ports were optimized to facilitate the extraction of a hydrogel biopsy from the collagen hydrogel using regular micropipette tips (Fig. 5a, b). After collagen biopsy digestion, cells were released from the hydrogel, retaining >90% cell viability (Fig. 5c–e). The cavity left in the hydrogel after the biopsy can be replenish with various cell types or hydrogel compositions. This property allowed us to manipulate tumor density, from low-density regions where no hypoxia is observed, to high-density tumors where cell metabolism spontaneously generated a hypoxic core inside of the microdevice. In this context, we seeded HCT-116 cells at a low cell density (1.5 million cells/mL) which did not lead to hypoxic conditions; therefore mimicking the non-hypoxic periphery regions of a tumor mass. Then, we used the ports of our platform to obtain a guided biopsy and re-filled the cavity with a collagen mixture containing a high density of HCT-116 cells (30 million cells/mL). We used a hypoxia-sensing reagent to reveal a hypoxic core corresponding to the area of high cell density Fig. 5f.

**Genetic response to nutrient Gradients.** Our previous experiments in this paper showed the capacity of the microdevice to generate gradients of viability in response to nutrient starvation. We next studied how these reconfigurable nutrient gradients impacted gene expression in HCT-116 cells. We seeded HCT-116 cells at a high density, added media/PBS following the necrotic front configuration (Fig. 6a) and after 24 h, we isolated biopsies of

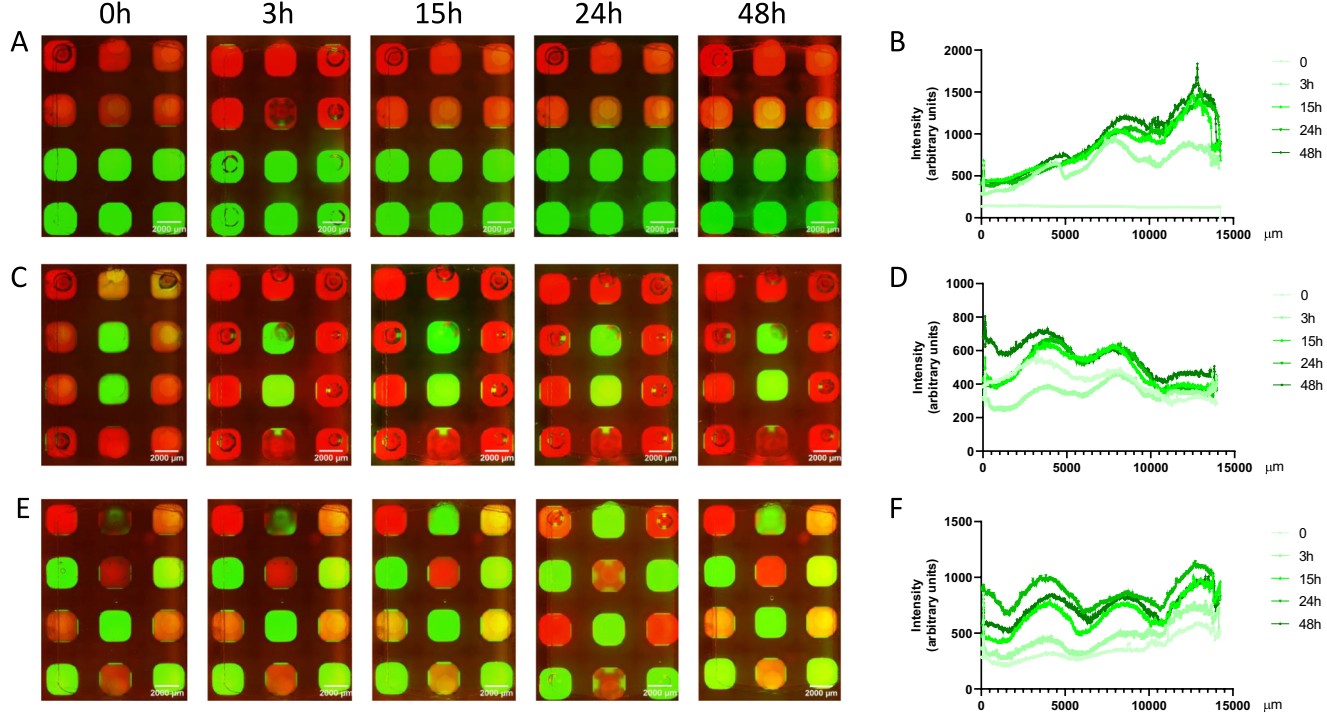

**Fig. 2 Dextran diffusion.** Representative images of 40 kDa FITC-Dextran and 70 kDa TR-dextran diffusion in the Griddient in lineal (**A**), radial (**C**), or periodic (**E**) configuration at different time points. Quantification of FITC-dextran intensity across the griddient over the time for lineal (**B**), radial (**D**) and periodic (**F**) configurations. $n = 3$.

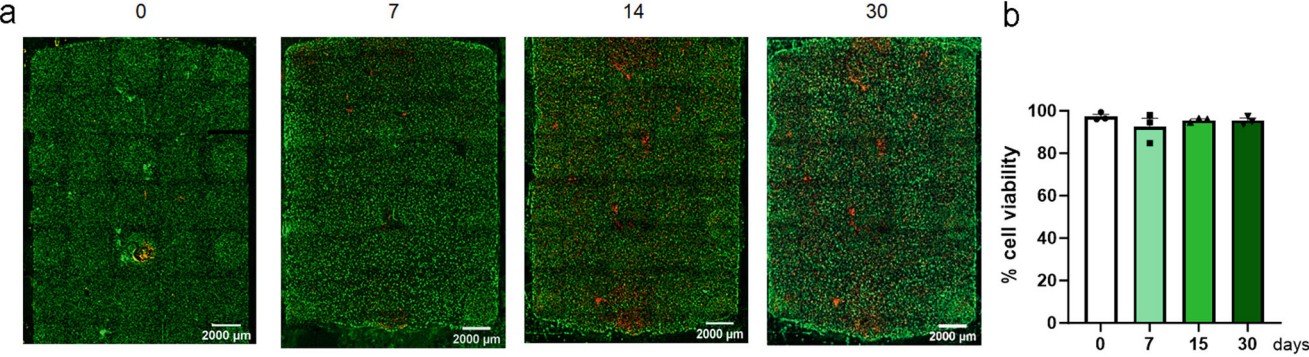

**Fig. 3 Cell viability in Griddient platform over time.** HCT-116 cells were stained with CellTracker™ Green CMFDA and seeded at 30 million cells/mL in the collagen hydrogel. Dead cells were stained with IP and image at 0, 7, 14, and 30 days. **a** Representative fluorescence images of each time point. **b** Cell viability quantification. Results are plotted as (mean ± SEM; $n = 3$ from two independent experiments).

the cells from several locations within the microdevice chamber. We observed that genes involved in various cell functions were affected by nutrient starvation (Fig. 6b, c). The results demonstrated that cells located further from the nutrient source upregulated genes involved in apoptosis (e.g., *CASP7*) and stress response (e.g., *NFE2L2*). Genes involved in glucose metabolism (e.g., *SCL1A1*, *PFKB3*, *PDK3*, *CA9*) were increasingly downregulated the farther they were from the media zone, suggesting a potential adaptive metabolic switch caused by the nutrient-depleted environment. Additionally, some genes exhibited a linear and progressive kinetic (e.g., *PCK1*, *BCL2*) whereas others seemed to plateau after a given distance from the nutrient source (e.g., *CASP7*), highlighting the potential of the platform to monitor different spatial dynamics of genetic changes.

**Metabolic plasticity.** The gene expression analysis demonstrated that HCT-116 cells exposed to nutrient starvation exhibited a different molecular phenotype compared to that of HCT-116 cells located under the source of nutrients (Figs. 6 and 7a). Previous studies have shown that cancer cells adapt their metabolism based on the nutrient-availability in the surrounding microenvironment. We generated transient gradients in Griddient to study how tumor cells changed their metabolism in real time. We used two-photon (2 P) optical metabolic imaging (OMI), a non-destructive, label-free method to analyze the natural auto-fluorescence properties of NAD(P)H (reduced form of nicotinamide adenine dinucleotide (phosphate)) and FAD (flavin adenine dinucleotide), metabolites involved in cell metabolism and redox ratio due to their roles as an electron donor and acceptor, respectively. We seeded HCT-116 at high density in the Griddient and cultured them for 12 h with media in all reservoir wells. Next, we substituted the media for PBS in half of the reservoir wells for 24 h to create a nutrient-depleted environment (Fig. 7a). Finally, we switched to fresh media again for 24 additional hours. We

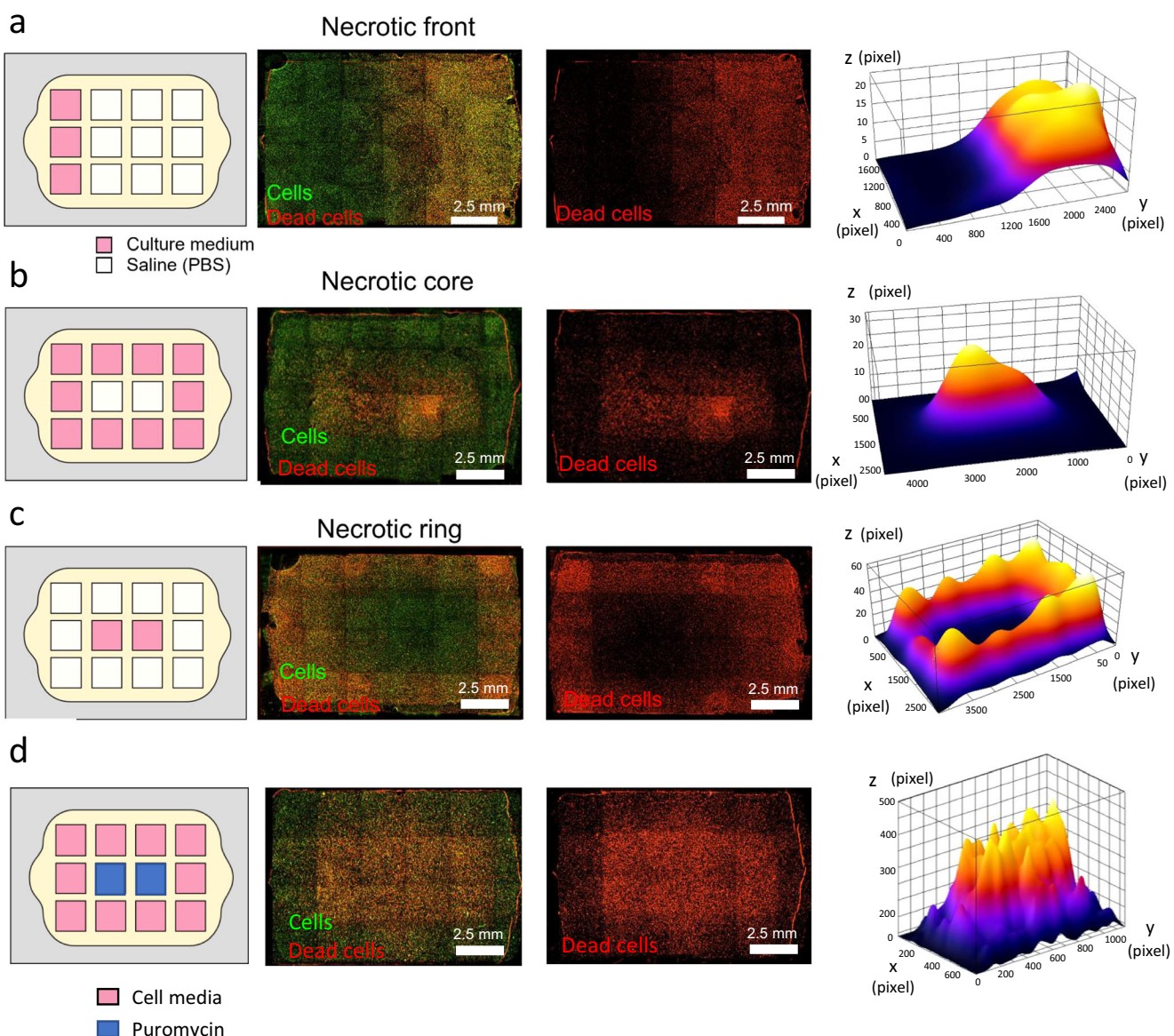

**Fig. 4 Controlling nutrient starvation and cell viability in the Griddient.** HCT-116 cells were stained with CellTracker™ Green CMFDA and seed at 30 million cells/mL. **a–c** Fluorescence images of the Griddient with different media/PBS configuration. Media was replaced by PBS as represented in the left panels. After 24 h cell viability was evaluated staining dead cells with PI. Images show the overlay between total cells and dead cells. Graph shows the fluorescence intensity profile of necrotic cells across Griddient $n = 3$. **d** Center wells of Griddient were treated with puromycin 5 µg/mL. After 24 h, cell viability was evaluated staining death cells with PI. Images show the overlay of viable and dead cells. Graph shows the fluorescence intensity profile of necrotic cells across the Griddient platform.

used OMI to monitor HCT-116 cell metabolism throughout this process, allowing us to observe metabolic changes in cells cultured in nutrient-depleted (i.e., PBS) or nutrient-enriched environments (i.e., culture medium). This approach also allowed us to monitor metabolic changes after the nutrient source was restored. OMI analysis revealed that HCT-116 cells located under the nutrient source exhibited a similar optical redox ratio (ORR), defined as the NAD(P)H intensity divided by the sum of the NAD(P)H and FAD intensities, at all-time points, suggesting that nutrient concentration was sufficient to maintain redox balance. Nutrient starvation caused a significantly more oxidized ORR in those cells cultured in nutrient-depleted environments (i.e., PBS). Interestingly, cells cultured at the interface between nutrient-enriched and nutrient-depleted environments showed a moderate change in their ORR (Fig. 7b, c). After restoring the nutrient concentration, cells in the depleted regions did not recover their

original ORR. However, cells seeded in the interface demonstrated a partial recovery after media was restored, exhibiting an ORR slightly more oxidized compared to cells cultured in nutrient-enriched environments (Fig. 7c). The mean fluorescence lifetime of NAD(P)H ($\tau_m$), which captures the protein-binding activities of NAD(P)H[16] showed similar trends to the ORR (Fig. 7d, e). These observations highlight the capacity of cancer cells to display transient metabolic adaptations that may have long-lasting effects on cell metabolism, further emphasizing the importance of metabolic reprogramming under physiological and pathological conditions.

**Culture of cancer organoids in the Griddient plate for immunotherapy research.** The Griddient device could offer multiple applications for a variety of research fields such a angiogenesis, tumor biology, or immunology. To explore these

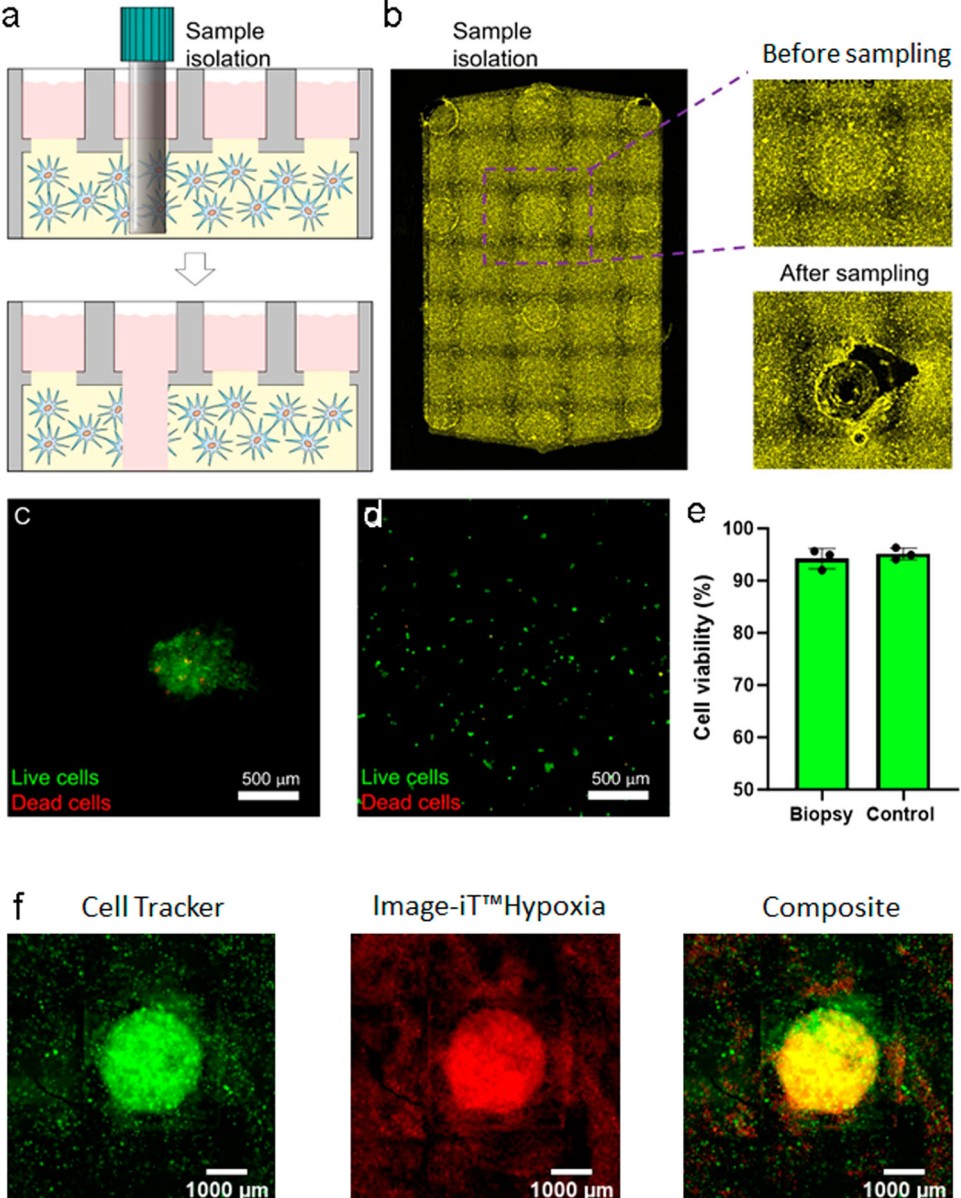

**Fig. 5 Generation of biopsy and grafts in Griddient platform. a** schematic representation of sample biopsy from one port in the Griddient platform. **b** HCT-116 cells were stained with CellTracker™ Deep Red Dye and seeded in collagen at 30 million cells/well. After gel polymerization, a biopsy was taken from one of the ports. Fluorescence images of the gel before and after biopsy. **c–e** The biopsy sample was stained with calcein (green, viable cells) and PI (red, dead cells). Cell images before (**c**) and after (**d**) biopsy digestion. **e** Quantification of cell viability in the biopsy.). Results are plotted as mean ± SEM; $n = 4$. HCT-116 cells plated in a 2D culture plate were used as a control. **f** First, a biopsy was performed in the gel. The cavity was filled with a collagen solution with higher cell density. Cells were stained with Cell Tracker (green) and ImageiT Hypoxia Reagent (red).

applications, we performed several proof-of-concept experiments (Fig. 8).

Blood vessels play a pivotal role in nutrient availability by controlling metabolite diffusion through the endothelial wall. We set out to explore the capacity of the Griddient plate to establish an in vitro model of endothelial barrier permeability (Fig. 8a, upper panels). After polymerizing the collagen hydrogel, we seeded HUVECs at a density of 20 million/mL on the reservoir wells. Following 24 h of cell culture, HUVECs were attached to the collagen interface, forming a monolayer in the interface with the media (Fig. 8a, bottom left panel). We decided to evaluate the capacity of the endothelial monolayer to respond to inflammatory signals, which, in vivo, decrease vascular integrity and metabolite and cellular permeability. Once we established a HUVEC

monolayer, we exposed the cells to 20 ng/mL TNF-α for 24 h. As expected, treatment with TNF-α induced the generation of pores in the endothelium as well as modifications in cell morphology. These effects of TNF-α have been already described in the literature[17]. (Fig. 8a, bottom right panel).

Interactions between tumor, stroma, and immune systems play a critical role in the context of cancer progression and treatment response. Cancer metastasis leads to tumor spread to other organs (e.g., bone marrow) where malignant cells must adapt and grow to a different environment compared with the tissue of origin (e.g., colon). As clinicians continue to explore new treatments such as immunotherapy, we set out to explore the potential of the Griddient in modelling metastasis. We generated a model of bone marrow metastasis where we co-cultured HCT-116 cancer

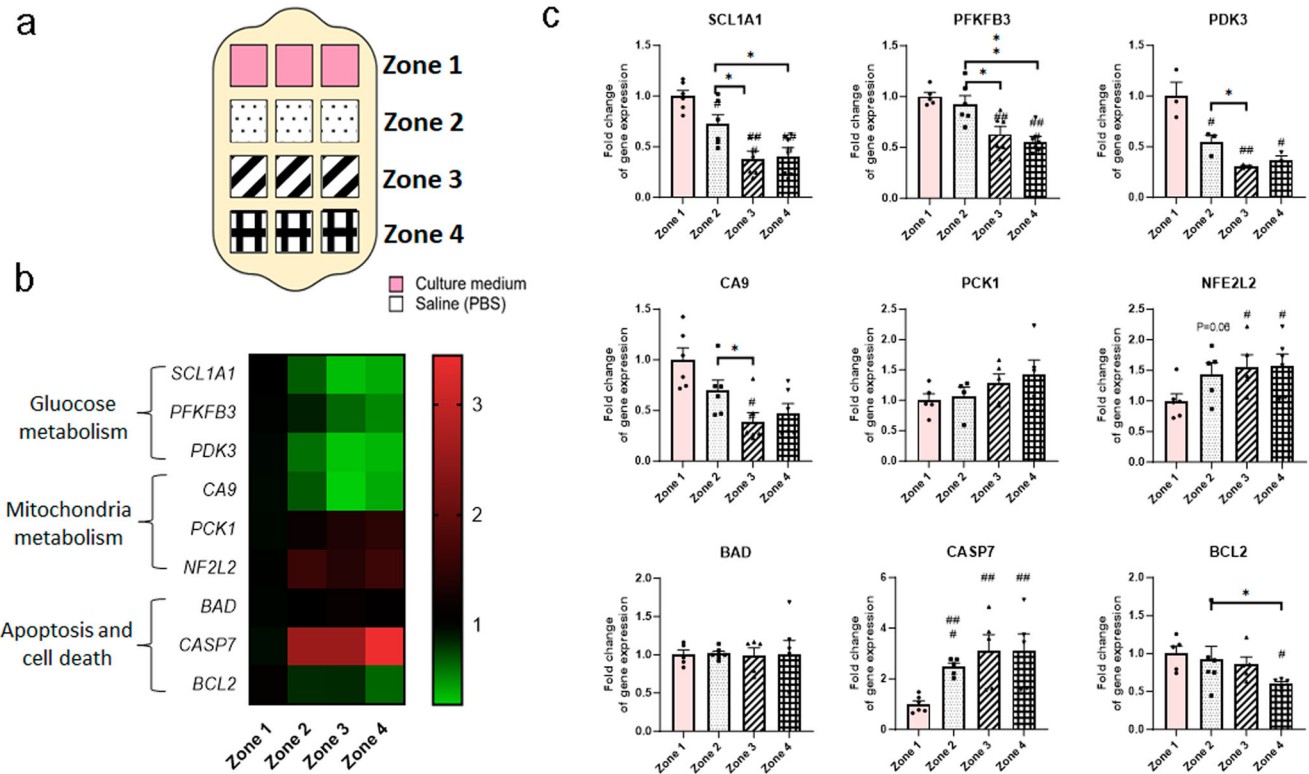

**Fig. 6 Cells modify gene expression in response to nutrient gradients.** HCT-116 cells were seeded at 30 million cells/mL in a collagen gel. **a** Spatial configuration of culture media/PBS distribution in the Griddient. **b** Cluster diagram of analyzed genes obtained from 4 different zones in the Griddient platform. **c** Gene expression analysis of metabolic and apoptotic genes. mRNA levels were quantified by qRT-PCR, normalized by TBP and GAPDH as housekeeping genes. Results were analyzed as expression relative to HCT-116 cells cultured under media (zone 1). Results are plotted as 2-$\Delta\Delta$Ct (mean ± SEM; $n = 6$). $^*p < 0.05$, $^{**}p < 0.01$, and $^{***}p < 0.001$ using One-Way ANOVA.

spheroids in a collagen hydrogel that also included Human Mesenchymal Stem Cells (hMSC) isolated from bone marrow aspirates (Fig. 8b). After 4 days of cell culture, PI was added to the media in order to visualize dead cells. Microscopy images showed that both cell types remained viable and stable for at least 4 days in culture and hydrogel injection into the platform did not affect spheroid integrity (Fig. 7b).

We then loaded natural killer (NK) cells into the center reservoir port to study immune cell extravasation, and migration in the environment created within Griddient (Fig. 8c). We observed that NK cells were able to penetrate and migrate through the collagen hydrogel. After 4 days of cell culture, PI was added to the media in order to visualize dead cells. NK cells remained viable after 15 days in culture and continued migrating and proliferating into the Griddient chamber (Fig. 8c), highlighting the potential of the platform to study new therapies based on immune grafting (i.e., direct injection of immune cells in the affected tissue).

## Discussion

Molecular gradients are an important, ubiquitous, and evolutionarily conserved signaling mechanism for guiding cell growth, migration, and differentiation; playing an essential role in physiological and pathological phenomena including inflammation, wound healing, and cancer. Microfluidic technology has proven to be versatile in its ability to employ a broad range of methods for exposing cells to engineered gradients. However, most of these methods are complex (requiring highly specialized personnel and equipment), and struggle to provide dynamic control over the gradient profile or are low throughput[18]. The most common techniques for the generation of gradients include: the use of

biological hydrogels premixed with the molecule of interest[19,20] or the use of transwell[21], Zigmond[22] and Dunn[23] chambers. The transwell device has a porous membrane that devices two wells, creating a vertical gradient of the molecule of interest. In Zigmond chamber, cells grow and migrate on a coverslip glass, which is located between two connected reservoirs that create an horizontal gradient. Dunn chamber is similar to Zigmond chamber but includes an additional third reservoir. The main difficulties with these three devices are the lack of gradient stability and the inability to control or reverse the gradients, which prevent them from accurately reflecting the complex gradients shown in vivo[24]. The Griddient platform exhibits improved potential to manipulate the gradient profile on-demand. Moreover, The Griddient supports the generation of multiple gradients in the same platform, including two-dimensional gradients in the XY axis.

We have created a user-friendly platform capable of generating controllable and reversible gradients in a 3D microenvironment. Plasma treatment creates a hydrophilic surface that facilities collagen solution flow, reducing the possibilities of bubble formation. Moreover, the design of the chamber helps to maintain the integrity of the collagen gel while the gel/reservoir ratio ensures the stability of the gradient over the time. We observed that large differences in volume among the reservoir wells, generate an incidental pressure gradient that created a convective flow through the hydrogel, which in turn affects the diffusion profile. In this experiment, one well contained 70 μL of dextran solution while all the other well contained only 10 μL, creating a significant pressure difference. This property could be leveraged to accelerate the perfusion of the system with any molecule of interest. Our platform is compatible with the vast majority of techniques used in cell and molecular biology, including

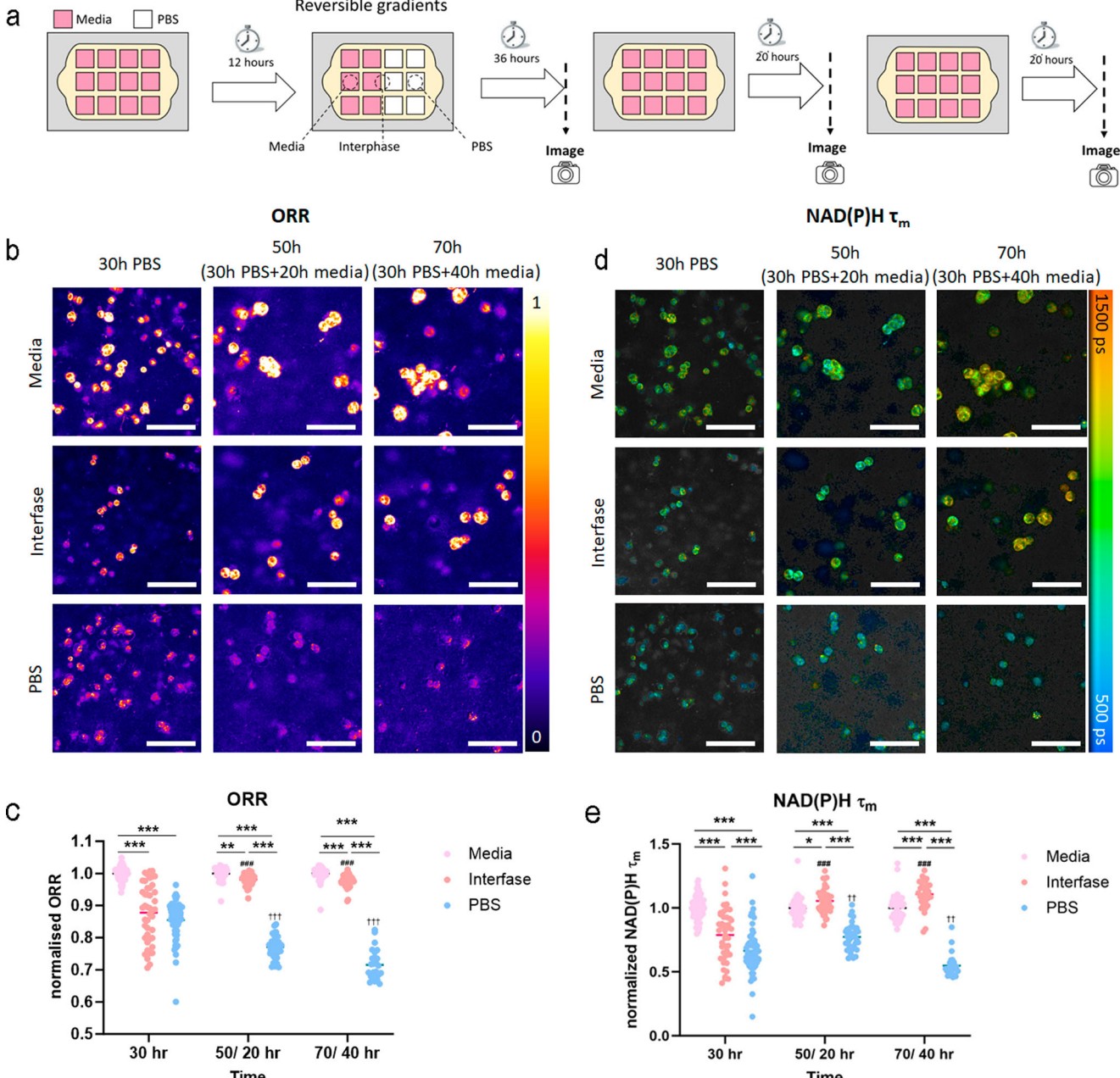

**Fig. 7 Cell metabolism in response to nutrient gradients. a** Schematic representation of the experimental settings. HCT-116 cells were seeded at 30 million cells/mL in a collagen hydrogel. After 12 h media was replaced by PBS following the described configuration. After 30 h, OMI images were obtained. Then PBS was replaced with media again. Cells were cultured under media for another 20 h (50 h post initial PBS change) before cells were imaged again. Cells were culture for 20 additional hours (70 and 40 h post initial PBS change and subsequent return to media). OMI images (**b**) and quantification (**c**) of the ORR (Optical Redox Ratio) after 30 min in PBS/media, and at 20 and 40 h after PBS was replaced by media. OMI images (**d**) and quantification (**e**) of of NAD(P)H $\tau_m$ (mean lifetime) after 30 min in PBS/media, and at 20 and 40 h after PBS was replaced by media. Significant main effects of different medias at the same time point are denotated by asterisks (*), significant comparisons between different time points are denotated by a hash (#) for the interface and a dagger (†) for PBS. For all symbols conferring statistical significance: single symbol $p < 0.05$, double symbol $p < 0.01$, triple symbol $p < 0.005$ using One-Way ANOVA. For each condition a total of at least 40 cells from two independent experiments were analyzed with 3 Griddient plates per experiment. Scale bars represents 100 μm.

fluorescence microscopy, metabolic imaging, and qPCR, requiring only conventional micropipettes.

We tested the Griddient device with a tumor model mimicking the three regions observed in solid tumors (necrotic core, quiescent middle layer, and proliferative perimeter). Unlike previous devices, our study demonstrates that Griddient has the capacity to generate diverse spatial configurations, dynamically modify

gradients over time, and investigate cellular plasticity[25]. These features make Griddient a promising tool for drug discovery and chemotherapy resistance assays. The capacity to create reversible drug gradients makes the Griddient a versatile platform to evaluate new therapeutic candidates targeting specific metabolic adaptations. Within solid tumors, drugs diffuse through the tumor mass, generating a concentration gradient. Sublethal drug

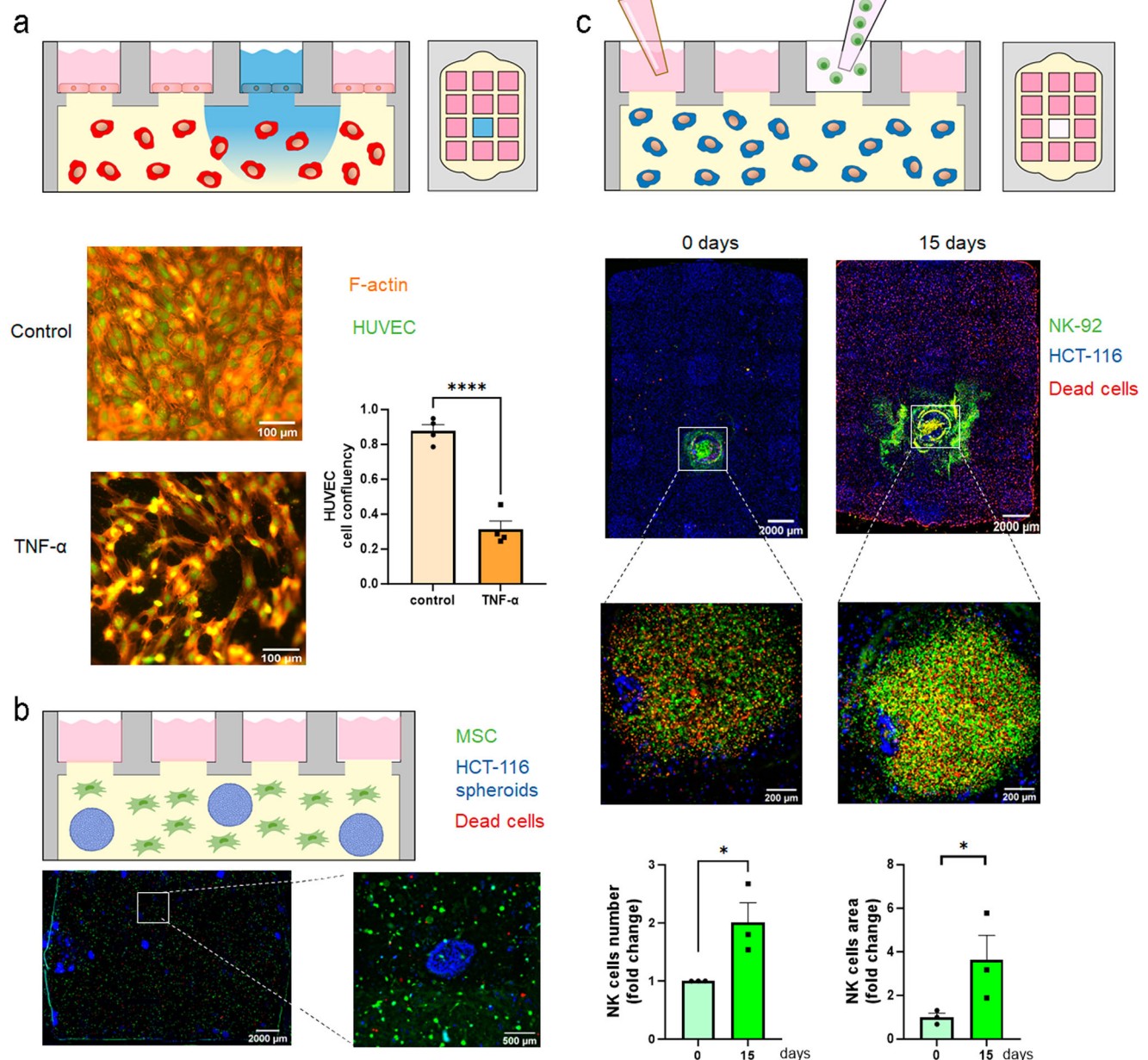

**Fig. 8 Different types of cell co-culture in the Griddient. a** Collagen gel was added in the main chamber. After polymerization, HUVEC cells, stained with CellTracker™ Green CMFDA, were seeded in the collagen interface. Cells were treated with vehicle or 20 ng/mL TNF-α for 24 h. Cells were stained with phalloidin and visualized on a fluorescence microscope. Area covered by cells was analyzed and plot as mean ± SEM, $n = 3$ **b**. HCT-116 spheroids (blue) were combined with Mesenchymal Stem Cells (MSC, green) in the collagen gel and seeded in the main chamber. IP was added to the media to visualize dead cells. After 24 h the Griddient Platform was imaged on a fluorescence microscope. **c** HCT-116 cells (blue) were seeded at 30 million cells/mL in a collagen solution. A biopsy of the gel was isolated, and the cavity was filled with a hydrogel containing NK-92 cells (green). IP was added into the media to visualize dead cells. Images was obtained in a fluorescence microscope at day 0 and 15 after the addition of NK-92 cells. Quantification of the NK cell number and area covered by NK cell at day 0 and 15. Data was normalized by area covered at day 0 and plot as mean ± SEM, $n = 3$.. For all symbols conferring statistical significance: single symbol $p < 0.05$, double symbol $p < 0.01$, triple symbol using Student T- test.

doses within the tumor might induce resistance. The Griddient allows us to easily study drug diffusion and evaluate the mechanisms underlying drug resistance.

Many cancers such as breast, prostate, lung and colon carcinomas have a higher tendency to migrate to the bone marrow. Once disseminated tumor cells reach the bone marrow, they enter a dormant phase to evade apoptosis induced by factors or chemotherapies. Although targeting bone metastasis pathways is the subject of several clinical trials[26–31], the underlying process remains poorly understood due to a lack of in vitro tools that can

generate complex environments while also being compatible with high-throughput models. Here we have reproduced a colon cancer bone metastasis that will be compatible with molecular studies or drug testing.

On the other hand, over the last decade, cell grafting has attracted the interest of companies focused on tissue implants for skin, cartilage, bone, and even brain[32]. Despite this, the number of in vitro devices available to study cell/tissue grafting remains limited. Our research has demonstrated the potential of Griddient to serve as a platform for cell grafting, since our platform

supports the generation of biopsies that can be further filled with a hydrogel of a different cellular content and composition.

## Materials and methods

**Microdevice design and fabrication**. Devices were machined on a CNC mill out of 384 polystyrene well-plates with a 0.02-inch end-mill. We milled out the first 400 μm of the well-plate bottom and then milled the hydrogel chamber (250 μm-depth) and the circular diffusion ports (1.7 mm diameter). Then, the platform was flipped upside-down and bonded to another 0.19 mm thick polystyrene layer using double-sided AR tape (ARcare 8890) with cutouts for the hydrogel chamber to ensure adequate sealing, while keeping optical transparency unchanged. Microdevices were treated with 100 W oxygen-enriched plasma for 2 min and then sterilized under UV light for 15 min prior to cell seeding.

**Cell culture**. The HCT-116 cell line was a kind gift from Dr. Dustin Deming (McArdle Laboratory for Cancer Research, University of Wisconsin-Madison) and cultured in RPMI 1640 (Thermo Fisher Scientific, 21870076) supplemented with 10% FBS (Thermo Fisher Scientific, 26140079).

HUVEC cells were obtained from ATCC (CRL-1730) and maintained in EGM-2 Endothelial Cell Growth Medium-2 BulletKit (Lonza, CC-3162).

To seed cells in Griddient, cells were first detached using a 0.05% trypsin/EDTA solution and resuspended at the desired density in PBS or RPMI 1640. Next, a type 1 rat tail collagen hydrogel (4.0 mg/mL) containing HCT-116 cells was prepared as follows: 10 μl of 10× phosphate-buffered saline (PBS), 1.2 μl of 1 M NaOH, 47.5 μl of collagen type I (8.43 mg/mL), 31.3 μl of distilled water, and 100 μl of cell suspension. 90 μl collagen hydrogel was injected into each microchamber of the microfluidic device and polymerized at room temperature for 15–20 min. Then, 50 μl of medium or PBS was then added to each well, and the Griddient was kept in the incubator.

**Cell staining and live cell fluorescent indicators**. HCT-116 and HUVEC cells were labeled with several fluorescent markers: Vybrant DiO (green; Thermo Fisher Scientific, V22886), or DiD (infra-red; Thermo Fisher Scientific, V22887), respectively. Briefly, cells were counted and subsequently suspended in PBS at 1 million cells/mL. Vybrant labeling agent was added at 5 μl/mL of cell suspension and incubated at 37 °C for 15 min. Cells were washed twice to remove excess dye before being embedded in collagen.

Cell viability was measured with propidium iodide (PI; final concentration of 2 μg/mL) (Thermo Fisher Scientific, P1304MP), following manufacturer instructions, which was added to the collagen mixture prior to polymerization, and to the culture media.

**Optical metabolic imaging (OMI)**. Optical redox ratio values for all conditions were normalized to the control condition for the same position (proximal or distal). NAD(P)H and FAD lifetime images were analyzed using SPCImage software (Becker &Hickl, Berlin, Germany) as described previously[33,34]. The fluorescence lifetime decay curve was deconvolved with the instrument response function and fit to a two-component exponential decay model at each pixel, $I(t) = \alpha_1 * e^{(-t/\tau_1)} + \alpha_2 * e^{(-t/\tau_2)} + C$, where $I(t)$ represents the fluorescence intensity at time $t$ after the laser excitation pulse, $\alpha$ accounts for the fractional contribution from each component, $C$ represents the background light, and $\tau$ is the fluorescence lifetime of each component. Since NAD(P)H exists in two conformational states, free or enzyme-bound, a two-component model was used. For NAD(P)H, the short and long lifetime components correspond with the free and bound conformations, respectively. The mean lifetime ($\tau_m$) was calculated using $\tau_m = \alpha_1\tau_1 + \alpha_2\tau_2$. The ORR was determined from the NAD(P)H and FAD intensities, where for each pixel, the intensity of NAD(P)H was divided by the sum of the NAD(P)H and FAD intensities. Using Cell Profiler, an automated cell segmentation pipeline was created[34]. This system identified pixels belonging to nuclear regions by using a customized threshold code. Cells were recognized by propagating out from the nuclei within the image. To refine the propagation and to prevent it from continuing into background pixels, an Otsu Global threshold was used. The cell cytoplasm was defined as the cell borders minus the nucleus. Values for NAD(P)H $\tau_m$, NAD(P)H intensity, FAD intensity, and the ORR were averaged for all pixels within each cell cytoplasm. At least 100 cells per sample were analyzed. Cell viability microscopy images were analyzed using Fiji (https://imagej.net/Fiji/Downloads).

**Cell imaging and analysis**. Griddient was imaged using a Nikon Eclipse Ti inverted fluorescence microscope. Images were obtained at the central point of the chamber. The reference was maintained across different conditions. Image analysis was performed using ImageJ Fiji software.

**Statistical analysis and reproducibility**. All the experiments were repeated at least three times as independent biological replicates. All results are presented as the mean standard deviation. Data were analyzed using GraphPad Prism v9 and statistical significance was set at $p < 0.05$. Student's T-test or One-way ANOVA test was performed depending on the number of comparisons required. For each experiment, at least 3 independent experiments have been performed and analyzed.

**Reporting summary**. Further information on research design is available in the Nature Portfolio Reporting Summary linked to this article.

## Data availability

The experimental data and the simulation results that support the findings of this study are available in Figshare under the project https://figshare.com/projects/Griddient_a_microfluidic_array_to_generate_reconfigurable_gradients_on-demand_for_spatial_biology_applications/175710. The identifiers for every experiment are the following: 10.6084/m9.figshare.2396101536; 10.6084/m9.figshare.2396103337; 10.6084/m9.figshare.2396103938; 10.6084/m9.figshare.2396104539; 10.6084/m9.figshare.2396105740

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

## Acknowledgements

University of Wisconsin Carbone Cancer Center Cancer (AAB7173), Center Support Grant NIH P30CA014520, NIH NCATS UH3CA260692 and NIH NIAID R01AI134749 and SEED program (101-4-534300-AAK3854). Morgridge Research Institute.

## Author contributions

C.S-D, M.V-M. and B.H. conceived, planned, and carried out the experiments. T.D.J and D.S.J. contributed in the design and optimization of Griddient fabrication procedures. C.A.R-M. and S.A-C. fabricated the Griddient platforms. J.R., E.C.G. and M.C.S. analyzed the Optical Metabolic Imaging data. J.P. and N.J.H. helped with sample preparation. J.M.A and D.J.B. conceived the original idea. J.M.A. supervised the project. All authors discussed the results and contributed to the final manuscript.

## Competing interests

D.J.B. holds equity in Bellbrook Labs LLC, Tasso Inc., Turba LLC, Salus Discovery LLC, Stacks to the Future LLC, Lynx Biosciences, Inc. and Onexio Biosystems LLC.
