## [Peer Review File · Communications Biology]

Reviewers' comments:

Reviewer #1 (Remarks to the Author):

This manuscript introduces a user-friendly platform that enables to generate controllable and reversible biochemical gradients in a 3D microenvironment. This is very relevant for studying the dependency of biological processes on such gradients.

The design is clever: 32 microfluidic chambers are connected by diffusion ports to arrays of reservoir wells in 384 well plate format. The paper convincingly demonstrates gradient formation (in various patterns) and cell culture capabilities. A strong feature of the platform is that cells can be retrieved for downstream analysis.

The platform is demonstrated with biologically relevant tests using cancer cells, endothelial cells, cancer spheroids and immune cells, which nicely highlight the possibilities of the platform.

The manuscript is clearly written, the platform is novel, and the experimental results are convincing. The platform should be interesting for researchers working in life sciences broadly, and in vitro testing in particular. I advise to accept the manuscript for publication in Communications Biology.

Reviewer #2 (Remarks to the Author):

This study presents a novel microfluidic system for generating biochemical gradients and spatial signals in 3D hydrogels. Multiple benchmarking characterizations were performed, including cell viability, metabolic activity, signaling, and co-culture activity. Overall the authors have developed a simple-to-use, intuitive, and highly adaptable platform that should be of broad interest to the cell and tissue biology communities. Some comments to consider:

1. Please indicate length scales for the device schematics in Figure 1.
2. In Figure 3, is the right panel a 3D intensity profile or is the z-coordinate intensity? Are these based on projected images or a single z-focal plane (and if so, which plane and how was it chosen)?
3. In Figure 6, "interphase" should be "interface".
4. Some quantitative characterizations should be performed on the gradient spatial profiles over time (e.g. from Figure 1). How quickly does the gradient establish and how stable is it over time? Both diffusion and any incidental pressure gradients (from volume mismatch at the media chambers, which can be caused by pipetting errors, evaporation, etc.) will lead to altered/perturbed gradient profiles over time. How significant are these effects? Is there an optimal/practical time window during which the gradient is most stable? Are there optimal practices (e.g. refilling media reservoirs at appropriate time points, amount of media to fill each reservoir, etc.) that can most effectively stabilize the gradient profiles over long term culture?
5. More details should be provided on the statistics, including the number of sample points and independent experiments per condition, for each analysis plot.
6. Some characterizations should be performed for Figure 7.
7. Over long-term, cells can spread to the top or bottom (or side) surfaces of the device, which can change their behaviors and morphologies (this appears to be the case in Fig. 7A). This could impact signaling and metabolic profiles. Some comments/discussions should be made.
8. Collagen gels can compact over time, which can generate spatial heterogeneities of the collagen substrate (e.g. gel edges can regress from the device boundaries). Are these observed? Some discussions should be made.
9. Do bubbles form during collagen filling into each well, and how is this addressed?

10. Some more discussions on comparison of the Griddient device with other gradient methods along with limitations of the Griddient device would be useful.

Reviewers' comments:

Reviewer #1 (Remarks to the Author):

This manuscript introduces a user-friendly platform that enables to generate controllable and reversible biochemical gradients in a 3D microenvironment. This is very relevant for studying the dependency of biological processes on such gradients.

The design is clever: 32 microfluidic chambers are connected by diffusion ports to arrays of reservoir wells in 384 well plate format. The paper convincingly demonstrates gradient formation (in various patterns) and cell culture capabilities. A strong feature of the platform is that cells can be retrieved for downstream analysis.

The platform is demonstrated with biologically relevant tests using cancer cells, endothelial cells, cancer spheroids and immune cells, which nicely highlight the possibilities of the platform.

The manuscript is clearly written, the platform is novel, and the experimental results are convincing. The platform should be interesting for researchers working in life sciences broadly, and in vitro testing in particular. I advise to accept the manuscript for publication in Communications Biology.

We thank the reviewer for their positive comments.

Reviewer #2 (Remarks to the Author):

This study presents a novel microfluidic system for generating biochemical gradients and spatial signals in 3D hydrogels. Multiple benchmarking characterizations were performed, including cell viability, metabolic activity, signaling, and co-culture activity. Overall the authors have developed a simple-to-use, intuitive, and highly adaptable platform that should be of broad interest to the cell and tissue biology communities. Some comments to consider:

1. Please indicate length scales for the device schematics in Figure 1.

We have included scale bars in Figure 1. We have also included an additional Supplementary Figure that includes detailed schematics with all the dimensions for the device.

2. In Figure 3, is the right panel a 3D intensity profile or is the z-coordinate intensity? Are these based on projected images or a single z-focal plane (and if so, which plane and how was it chosen)?

We appreciate the reviewer clarification. The Z-coordinate represents the fluorescence intensity obtained from the fluorescence microscopy images. For the images included in Figure 3, we chose the middle focal plane within the chamber. We kept that focal plane across multiple conditions. We also checked that cell viability remained constant across the Z-axis (the hydrogel height is only 250-300 μ m while X and Y dimensions are in the range of 1 cm).

We believe it is worth mentioning that we used fluorescence microscopy, which can't discriminate well across the Z-direction (as opposed to confocal microscopy), thus the microscopy images show several z-planes above and below the visualize plane. We have included this information in the materials and methods section (page 24).

3. In Figure 6, "interphase" should be "interface".

We appreciate the correction. We have modified the figure and the text accordingly.

4. Some quantitative characterizations should be performed on the gradient spatial profiles over time (e.g. from Figure 1). How quickly does the gradient establish and how stable is it over time? Both diffusion and any incidental pressure gradients (from volume mismatch at the media chambers, which can be caused by pipetting errors, evaporation, etc.) will lead to altered/perturbed gradient profiles over time. How significant are these effects? Is there an optimal/practical time window during which the gradient is most stable? Are there optimal practices (e.g. refilling media reservoirs at appropriate time points, amount of media to fill each reservoir, etc.) that can most effectively stabilize the gradient profiles over long term culture?

We appreciate the reviewer comment. We have included a new figure showing the diffusion of small molecules over the time. We have observed that after 15 hours the gradient profile remains mostly unchanged (compared with the results observed at 48 hours), suggesting the system is close to reaching the steady-state. These experiments were performed in a collagen gel without cells. If cells were added, the molecular gradient might be affected depending on cell consumption of such molecules. Additionally, we have examined the effect of incidental pressure gradients in molecule diffusion (Supplementary figure 2). We have added this information in the discussion section (page 26).

5. More details should be provided on the statistics, including the number of sample points and independent experiments per condition, for each analysis plot.

We agree with the reviewer. We have included that information in the figure legend of each plot.

6. Some characterizations should be performed for Figure 7.

Following the reviewer's suggestion, we have:

- 1. Calculated the cell confluency on Figure 7A.*
- 2. Analyzed the migration of NK cells on figure 7C.*

7. Over long-term, cells can spread to the top or bottom (or side) surfaces of the device, which can change their behaviors and morphologies (this appears to be the case in Fig. 7A). This could impact signaling and metabolic profiles. Some comments/discussions should be made.

We appreciate the reviewer's comment. In the case of Figure 7A we did not see migration of endothelial cells inside the hydrogel. TNF- α induced the formation of pores in the monolayer, as well as modifications in cell morphology. This effects of TNF- α have been already described in the literature (Kunimura et al., 2021). Regarding the possibility if tumor cells moving to the top/bottom/sides of the device, we selected HCT-116 cells for these experiments for their low migratory capacity. However, the reviewer is right that depending on the migration potential of the cells used, these effects may play a role. We have added this information in the main text (page 21)

8. Collagen gels can compact over time, which can generate spatial heterogeneities of the collagen substrate (e.g. gel edges can regress from the device boundaries). Are these observed? Some discussions should be made.

We cultured the Griddient with HCT-116 over 30 days and we did not see any collagen contraction, edge regression, or spatial heterogeneities. We suspect this behavior depends on the cell type/s included in the hydrogel (fibroblasts are known for their capacity to contract hydrogels). We have included this appreciation in the result and discussion sections (page 9 and 24).

9. Do bubbles form during collagen filling into each well, and how is this addressed?

We plasma treated the device before its use in cell culture, creating a hydrophilic surface that facilitates collagen solution flow, reducing the possibilities of bubble formation. We included this information in the discussion section (page 24).

10. Some more discussions on comparison of the Griddient device with other gradient methods along with limitations of the Griddient device would be useful.

We agree with the reviewer's suggestion. We have added additional microfluidic and non-microfluidic gradient methods in the first paragraph of the discussion section. We have also included a new paragraph discussing the limitations of the Griddient device (page 25-26).